# Improvement of Dynamic Performance and Detectivity in Near-Infrared Colloidal Quantum Dot Photodetectors by Incorporating Conjugated Polymers

**DOI:** 10.3390/molecules27217660

**Published:** 2022-11-07

**Authors:** Myeong In Kim, Jinhyeon Kang, Jaehee Park, WonJo Jeong, Junho Kim, Sanggyu Yim, In Hwan Jung

**Affiliations:** 1Department of Organic and Nano Engineering, Human-Tech Convergence Program, Hanyang University, 222 Wangsimni-ro, Seongdong-gu, Seoul 04763, Korea; 2Department of Energy Engineering, Hanyang University, 222 Wangsimni-ro, Seongdong-gu, Seoul 04763, Korea; 3Department of Chemistry, Kookmin University, 77 Jeongneung-ro, Seongbuk-gu, Seoul 02707, Korea

**Keywords:** quantum-dot photodetector, conjugated polymers, hole-transporting polymers, dark current suppression, dynamic properties, near-infrared detection

## Abstract

Colloidal quantum dots (CQDs) have a unique advantage in realizing near-infrared (NIR) photodetection since their optical properties are readily tuned by the particle size, but CQD-based photodetectors (QPDs) presently show a high dark current density (*J*_d_) and insufficient dynamic characteristics. To overcome these two problems, we synthesized and introduced two types of conjugated polymers (CPs) by replacing the *p*-type CQD layer in the QPDs. The low dielectric constant and insulating properties of CPs under dark conditions effectively suppressed the *J*_d_ in the QPDs. In addition, the energy-level alignment and high-hole mobility of the CPs facilitated hole transport. Therefore, both the responsivity and specific detectivity were highly enhanced in the CP-based QPDs. Notably, the dynamic characteristics of the QPDs, such as the −3 dB cut-off frequency and rising/falling response times, were significantly improved in the CP-based QPDs owing to the sizable molecular ordering and fast hole transport of the CP in the film state as well as the low trap density, well-aligned energy levels, and good interfacial contact in the CP-based devices.

## 1. Introduction

The detection of near-infrared (NIR) light is currently evaluated as an important technology in a wide range of fields, such as bioimaging, optical communications, and autonomous driving sensors [1,2,3,4,5,6,7,8,9]. Therefore, NIR photodetectors (PDs) that convert NIR light into electrical signals have been actively studied, resulting in the development of various types of active materials based on silicon, InGaAs (III–V compounds), organic dyes, and colloidal quantum dots (CQDs) [10,11,12,13,14,15]. Among these, CQDs are regarded as promising candidates for flexible NIR PDs. They can be deposited by a solution process under ambient conditions, and their bandgaps and energy levels can be easily tuned by the particle size, ligand exchange treatment, and/or surface modification [16,17,18,19,20,21]. Wei et al. reported lateral bilayer CQD PDs (QPDs) composed of PbS CQDs and *p*-*n* bulk heterojunction (BHJ) organic layers. Due to the BHJ organic layer, the dark current density (*J*_d_) value was decreased and the photoinduced current density (*J*_ph_) value was increased, resulting in an enhanced responsivity (*R*) of 0.25 A/W and a specific detectivity (*D**) of 2 × 10^12^ Jones under 950 nm light illumination of 10.6 µW/cm^2^ at a 40 V bias [22]. Zhao et al. reported perovskite:PbS hybrid QPDs using an antisolvent additive solution process, which evenly dispersed the PbS CQDs between the grain boundaries of the perovskite crystals. Owing to the well-modulated film morphology and perovskite crystals, the fabricated photodetector exhibited a high *D** value exceeding 10^12^ and 10^11^ Jones under visible and NIR illumination, respectively, at a −0.5 V bias [23]. Zhang et al. developed bilayer QPDs composed of PbS CQDs and perovskites. Due to the broad absorption covering the NIR region, the QPDs showed a high *R* value of 1.58 A/W and a *D** value of 3.0 × 10^11^ Jones under 940 nm illumination, but exhibited a relatively slow signal response time of 42 ms [24]. Moreover, our research group reported an improved QPD performance by introducing conjugated polyelectrolytes onto the electron transporting layer (ETL) to facilitate electron transport and suppress *J*_D_ at the interface between the CQD layer and ETL. The promising *D** values of 2.5 × 10^12^ and 1.3 × 10^12^ Jones were achieved under 532 nm green [25] and 940 nm NIR LED illumination [26], respectively, at −1 V.

In the present study, we synthesized two types of conjugated polymers (CPs), PSBOTz and PBB, to improve the NIR photodetection properties of QPDs. PbS CQDs are commonly synthesized by a single ligand exchange process using tetrabutylammonium iodide (TBAI), but they exhibit relatively high *J*_d_ values, low on/off ratios, and slow response times in devices [27]. To overcome these limitations, Chuang et al. developed electron-blocking and hole-transporting PbS CQDs via a ligand exchange reaction using 1, 2-ethanedithiol (EDT) [28], and the EDT-treated PbS CQDs significantly enhanced the charge collection efficiency by modifying the electric field and energy level of the PbS CQDs [29,30,31]. However, we found that the EDT-treated PbS CQDs (PbS-EDT) still exhibited substantial *J*_d_ and slow signal response times in the QPDs. In addition, the layer-by-layer process required multiple coating steps to deposit the PbS-EDTs. [32] In contrast, thiophene- and benzene-ring-based semiconducting polymers naturally possess *p*-type characteristics, and a uniform thin film is readily achieved by a simple spin-coating process owing to the high viscosity of the polymer solution [33,34,35,36]. In addition, several CPs were successfully utilized as a hole-transporting layer in the photovoltaic applications [37,38]. To understand the effect of CP on the QPDs, the CP-based QPDs were directly compared with the control PbS-EDT-based devices. The low dielectric constant and insulating properties of the CPs under dark conditions significantly reduced *J*_d_ in the QPDs [39], resulting in *D** values one order of magnitude higher than those of the PbS-EDT-based devices. In addition, the tunable highest occupied molecular orbital (HOMO) energy levels of CPs could control the hole-transporting properties by matching the energy level with the photoconductive TBAI-treated *n*-type PbS CQD (PbS-TBAI) layer, which improved the hole mobility and *J*_ph_ in the QPDs. Notably, the PBB polymer showing strong molecular ordering and the lowest trap density in the QPDs exhibited superior dynamic characteristics with a 9-fold increase in the frequency response and a 3–4-fold increase in the signal response speed in the devices. The utilization of CPs with high-hole mobility, low-trap density, and proper HOMO energy levels appears promising for high-performance QPDs

## 2. Results and Discussion

### 2.1. Material Characterization

PbS CQDs were synthesized via a modified hot-injection method using lead acetate trihydrate as the lead source [40]. The X-ray diffraction (XRD) spectrum of the synthesized PbS CQDs was measured and shown in Appendix A. The diffraction peaks of PbS CQDs appeared at 25, 31, 42, 51, 61, 69, and 71°, which corresponded to (111), (200), (220), (311), (400), (331), and (420) crystal lattices, respectively.

Appendix A shows the first excitonic peak of the synthesized PbS CQDs, and their optical bandgap ($E_{g}^{\mathrm{opt}}$) was calculated using Equation (1):(1)$$E_{g}^{\mathrm{opt}}=\frac{hc}{\lambda_{\max}}\;\;$$
where *h* is the Planck constant, *c* is the velocity of the light, and $\lambda_{\max}$ is the first excitonic peak. The measured $E_{g}^{\mathrm{opt}}$ from the $\lambda_{\max}$ at 867 nm was 1.43 eV.

Since the bandgap (*E*_g_) of the CQDs depends on the particle size, the bandgap can also be calculated by measuring the size of the CQDs (Equation (2)) from the transmission electron microscopy (TEM) images of PbS CQDs [16]:(2)$$E_{g}=0.41+{\left(0.0252d^{2}+0.283d\right)}^{-1}$$

The measured diameter (*d*) of the PbS CQDs was approximately 2.76 nm, and the corresponding *E*_g_ was 1.43 eV. The particle sizes of the PbS CQDs matched well with their optical bandgaps. The absorption properties and TEM images of the PbS CQDs are shown in Appendix A, respectively.

The conjugated polymers (CPs), PSBOTz and PBB, were synthesized via Stille polymerization according to our synthetic procedures [14,41] and their polymeric structures and ^1^H NMR were recorded in Figure 1 and Appendix A, respectively [14,42]. The number average molecular weights of the PSBOTz and PBB polymers were determined to be 5800 Da (*Đ* = 2.24) and 9400 Da (*Đ* = 2.06), respectively, by gel permeation chromatography (GPC) (Appendix A). The UV-Vis spectra of PSBOTz and PBB were compared in the film state (Figure 1a). Both the PSBOTz and PBB polymers showed similar absorption bands, and the absorption maxima in the film state were 520 and 524 nm, respectively. The optical bandgaps of the CPs were calculated from the onset wavelength of the film state. The onset wavelengths of PSBOTz and PBB were 616 and 628 nm, respectively, which corresponded to optical bandgaps of approximately 2.0 eV. FT-IR spectra of PSBOTz and PBB were shown in Appendix A. Both polymers showed C-H, C-C, C-N, and C-S stretching and C-H and C-C bending peaks in common, but PBB showed additional strong C-O stretching peaks at 1220–1183 cm^−1^ due to the existence of C-O bonding on the BBO moiety.

The HOMO energy levels of the synthesized materials were evaluated by cyclic voltammetry (CV), as shown in Figure 1b. All the CPs and PbS CQDs were coated onto ITO glass, which was used as the working electrode. The onset potentials of PSBOTz, PBB, PbS-TBAI, and PbS-EDT were 0.64, 0.81, 1.22, and 0.34 V, respectively, and the corresponding HOMO energy levels were −5.3, −5.5, −5.9, and −5.0 eV, respectively. An energy level diagram is shown in Figure 1c. All the hole-transporting materials (PSBOTz, PBB, and PbS-EDT) showed high-lying HOMO energy levels compared to PbS-TBAI, which indicates the formation of an efficient hole-transport pathway from the PbS-TBAI layer to the hole-transporting layer. However, the HOMO energy levels of PSBOTz (−5.3 eV) and PbS-EDT (−5.0 eV) were higher than those of the electron-blocking MoO_X_ layer (−5.3 eV), which does not seem to be ideal for efficient hole transport to the electrode. To utilize PbS-EDT as a hole-transporting layer, it is considered that an alternative electrode capable of better band alignment with the PbS-EDT is needed. In contrast, PBB had an intermediate HOMO energy level (−5.5 eV) located between the PbS-TBAI (−5.9 eV) and MoO_X_ (−5.3 eV) layers. Therefore, it is strongly expected that PBB had the most ideal HOMO energy level for hole transport in devices [37]. The optical and electrochemical properties of the CPs are summarized in Appendix A.

### 2.2. Static Properties of QPDs

NIR QPDs were fabricated with an *n-i-p* structure of ITO/ZnO/PbS-TBAI/PbS-EDT or CP/MoO_X_/Ag; the schematic device structure is illustrated in Figure 2a. The PbS-TBAI and PbS-EDT layers were obtained by a solid-state ligand exchange process using TBAI and EDT solutions, respectively. To obtain an appropriate thickness for the PbS CQD layer, the layer-by-layer spin-coating process was repeated five times for the PbS-TBAI layer and two times for the PbS-EDT layer. The CPs required only one spin-coating process owing to the sufficient viscosity of the polymer solution. The thickness of each layer was estimated from the cross-sectional SEM images, as shown in Appendix A.

The current density–voltage (*J*–*V*) curves were measured under the illumination of a collimated 940 nm NIR LED by changing the light intensity, as shown in Figure 2b–d. *R* is defined as the electrical output per optical input, which can be calculated using Equation (3), where *J*_ph_ is the photogenerated current density and *P*_light_ is the incident light power:(3)$$R\;\left(\frac{A}{W}\right)=\frac{J_{\mathrm{ph}}}{P_{\mathrm{light}}}\;$$

The *R* values of the PSBOTz, PBB, and PbS-EDT-based devices were 0.175, 0.220, and 0.217 A/W, respectively, under the illumination of 7 μW/cm^2^ at −1 V, and 0.101, 0.138, and 0.116 A/W, respectively, under the illumination of 5.0 mW/cm^2^ at −1 V. The PBB-based devices showed higher *R* values than the other devices. The well-aligned HOMO energy levels of PbS-TBAI (−5.9 eV), PBB (−5.4 eV), and MoOx (−5.3 eV) are expected to improve the hole transport properties of the devices.

*D** is the most important figure of merit for evaluating the photodetection performance, and it can be calculated from Equation (4), where *q* is the elementary charge and *R* is the responsivity, which was calculated from Equation (3):(4)$$D^{*}\left(\mathrm{cm}\;\frac{{\mathrm{Hz}}^{0.5}}{W},\;\mathrm{Jones}\right)=\frac{R}{\sqrt{2qJ_{d}}}\;\;$$

The *D** values of the PSBOTz, PBB, and PbS-EDT-based devices were 1.77 × 10^12^, 2.82 × 10^12^, and 8.24 × 10^11^ Jones, respectively, under the illumination of 7 μW/cm^2^ at −1 V, and 1.02 × 10^12^, 1.77 × 10^12^, and 4.39 × 10^11^ Jones, respectively, under the illumination of 5.0 mW/cm^2^ at −1 V. The CP-based devices exhibited significantly higher *D** values than the PbS-EDT-based devices due to the low level of *J*_d_ in the devices. This is a unique advantage of polymeric materials since the CPs act as insulators under dark conditions. However, PbS-EDTs are not good insulators under dark conduction since they contain mobile ions or ligands [43,44]. Therefore, PbS-EDT was limited to blocking *J*_d_ in the devices, whereas CPs can act as a *J*_d_ blocking layer under dark conditions. In particular, PBB-based devices showed the highest *D** values among the fabricated devices, which was attributed to the lowest *J*_d_ values and highest *R* values in the QPDs. The measured QPD properties were summarized in Table 1.

The hole mobility (*μ*_h_) was estimated from the space-charge-limited current (SCLC) of the hole-only device with an ITO/PEDOT: PSS/PbS-EDT or CP/MoO_X_/Ag structure, and the results are shown in Figure 2e. The *μ*_h_ was calculated via Mott-Gurney’s law using Equation (5):(5)$$J_{\mathrm{SCLC}}=\frac{9\varepsilon_{0}\varepsilon_{r}\mu_{\mathrm{SCLC}}V^{2}}{8L^{3}}\;$$
where $\varepsilon_{0}$ and $\varepsilon_{r}$ were the vacuum permittivity and relative dielectric constant, respectively. In this paper, the $\varepsilon_{r}$ value was set as 3.5 and 21.2 for the polymer HTL and PbS-EDT, respectively, which was the most widely used value. *L* was the thickness of active layer. The *μ*_h_ values of the PSBOTz, PBB, and PbS-EDT-based devices were 2.75 × 10^−3^, 9.63 × 10^−3^, and 1.77 × 10^−3^ cm^2^/V·S, respectively. PbS-EDT exhibited the lowest mobility among the three hole-transporting materials despite the high current density generation under the applied voltage owing to the high dielectric constant value of 21.2 [45]. In particular, the high dielectric constant of PbS-EDT facilitates *J*_d_ transport at the interfaces, which is closely related to the high *J*_d_ and low *D** in PbS-EDT-based QPDs [46]. In contrast, the PBB polymer exhibited the highest hole mobility with a low dielectric constant of 3.5, which could result in the best *R* and *D** values among the three hole-transporting materials.

To evaluate the charge generation rate under 940 nm LED illumination, the net photocurrent density–effective voltage (*J*_nph_ vs. *V*_eff_) was plotted, as shown in Figure 2g. The *J*_nph_ and *V*_eff_ values were calculated using *J*_nph_ = *J*_ph_ − *J*_d_ and *V*_eff_ = *V*_0_ − *V*, respectively, where *V*_0_ is the voltage when the *J*_nph_ value is 0. The maximum charge generation rate (*G*_max_) was obtained using Equation (6):*J*_sat_ = *qG*_max_*L*(6)
where *L* is the active layer thickness, *q* is the elementary charge, and *J*_sat_ is the saturated *J*_nph_ value from the *J*_nph_ vs. *V*_eff_ plot; the *J*_sat_ value was also set as the maximum *J*_nph_ value since there was a *J*_nph_ value that was not saturated [47]. From these equations, the calculated *G*_max_ values of the PSBOTz, PBB, and PbS-EDT-based devices were 1.13 × 10^26^, 3.66 × 10^26^, and 8.36 × 10^25^ m^−3^s^−1^, respectively. The PBB-based devices showed the highest *G*_max_ values, which implies that charge extraction was highly improved at the interfaces between the PBB and PbS-TBAI layers.

To evaluate the charge recombination properties of the QPDs, the dependence of *J*_ph_ and *V*_OC_ on *P*_light_ was investigated (Appendix A and Figure 2f). The relationship between *J*_ph_ and *P*_light_ is shown in Equation (7):(7)$$J_{\mathrm{ph}}\propto P_{\mathrm{light}}^{\alpha }\;$$

The α value is the empirical factor, and the bimolecular recombination in the devices is more efficiently suppressed as α approaches 1 [48]. The calculated α values of the PSBOTz, PBB, and PbS-EDT-based devices were 0.87, 0.94, and 0.88, respectively, under a bias of −1 V. The PBB-based devices exhibited the lowest bimolecular recombination, which is advantageous for achieving fast charge transport and efficient *J*_ph_ generation [49]. The open-circuit voltage vs. incident light (*V*_OC_ vs. *P*_light_) was plotted to estimate the trap-assisted recombination, as shown in Figure 2f:(8)$$V_{\mathrm{OC}}=\frac{nkT}{q}ln\left(P_{\mathrm{light}}^{\alpha }\right)+C\;$$
where *T* is the temperature, *q* is the elementary electron charge, *C* is a constant, *k* is Boltzmann’s constant, and *n* is the ideality factor. A *kT*/*q* value closer to 1 indicates a smaller trap-assisted recombination under *V*_OC_ conditions [50]. The calculated *n* values of the PSBOTz, PBB, and PbS-EDT-based devices were 1.45, 1.45, and 1.42, respectively, which indicates a similar degree of trap-assisted recombination in all three devices.

The trap densities of the PSBOTz, PBB, and PbS-EDT-based devices were investigated from the SCLC plots of the ITO/ZnO/PbS-EDT and CP/Al devices, as shown in Appendix A. The trap density was calculated using Equation (9) [51]:(9)$$V_{\mathrm{TFL}}=\frac{qNL^{2}}{2\varepsilon_{0}\varepsilon_{r}}\;$$
where *V*_TFL_ is the trap-filled limited voltage, which is set as the onset point where the slope is drastically increased, *L* is the thickness of the pristine film layer, ε_0_ is the vacuum permittivity, and ε*_r_* is the dielectric constant (PbS-EDT: 21.2 and CP: 3.5). The *N* value indicates the electron trap density, and the calculated values of PSBOTz, PBB, and PbS-EDT were 9.49 × 10^16^, 8.95 × 10^16^, and 1.28 × 10^17^ cm^−3^, respectively. The PBB-based devices exhibited the lowest trap densities among the three types of devices.

### 2.3. Dynamic Properties of QPDs

The dynamic characteristics of the NIR QPDs were evaluated from the transient photovoltage (TPV) response upon square-wave 940 nm wavelength irradiation in the self-powered mode, as shown in Figure 3b–d. The signal response time is defined as the time required for the detection signal to evolve from 10 to 90 % (rising time, *t*_r_) or from 90 to 10% (falling time, *t*_f_) of the steady-state voltage under a light intensity of 5.00 mW/cm^2^ [52]. As shown in Figure 3c,d, the *t*_r_ values of the PSBOTz, PBB, and PbS-EDT-based devices were 25, 14, and 33 μs, respectively. The PBB-based devices exhibited the fastest rising response among the three types of devices. Regarding *t*_f_, the signal responses of the PSBOTz-, PBB-, and PbS-EDT-based devices were delayed to 5.7, 2.6, and 12.6 ms, respectively, but the PBB-based devices still showed the fastest response time among the three types of devices. The highest hole mobility, lowest trap density, and well-aligned HOMO energy level of PBB resulted in the fastest signal response in the QPDs. Interestingly, the falling response speed of the PbS-EDT-based devices was extremely slow compared to the CP-based QPDs. In general, a long carrier recombination lifetime in photodiodes results in a slow response speed [53,54]. Therefore, the relatively long exciton lifetimes of CQDs could generate residual *J*_ph_ after removing the light sources, whereas the short exciton lifetime of CPs could effectively remove the off-light *J*_ph_ in the devices.

The −3 dB cut-off frequency (*f*_−3dB_) of the QPDs was measured using a collimated 940 nm LED with a power density of 5.0 mW/cm^2^ in the self-powered mode, as shown in Figure 3e. The measured *f*_−3dB_ values of the PSBOTz-, PBB-, and *p*-type PbS-based devices were 4.1, 19.3, and 2.2 kHz, respectively. Notably, the frequency response of the PBB-based devices was approximately one order of magnitude higher than the other devices. The frequency response also exhibited a tendency similar to the TPV responses.

The noise equivalent power (NEP) and linear dynamic range (LDR) were calculated using Equations (10) and (11), respectively, where *J*_max_ and *J*_min_ are the maximum and minimum current density values, respectively, which are in a linear relationship between *J*_ph_ and *P*_light_ [51].
(10)$$\mathrm{NEP}\;\left(\frac{W}{{\mathrm{cm}}^{2}}\right)=\frac{J_{d}}{R}$$
(11)$$\mathrm{LDR}\;\left(\mathrm{dB}\right)=20log\frac{J_{\max}}{J_{\min}}\;$$

The calculated NEP values of the PSBOTz-, PBB-, and PbS-EDT-based devices were 2.69 × 10^−7^, 1.67 × 10^−7^, and 1.41 × 10^−6^ W/cm^2^, respectively, at a bias of −1.0 V. The CPs were highly efficient in decreasing the NEP values owing to the low *J*_d_ values in the devices, and this low NEP is advantageous for detecting low-intensity light. The LDR values of the PSBOTz-, PBB-, and PbS-EDT-based devices were calculated to be 95, 107, and 88 dB at −1.0 V, respectively as shown in Appendix A [55]. The lower *J*_d_ values of the CP-based devices also resulted in a better LDR.

### 2.4. Morphological Characterization

The interfacial contact between the PbS-TBAI and CP layers is highly important for improving the charge extraction at their interfaces. To evaluate this, the water contact angles (θs) on the surfaces of the PbS-TBAI, PSBOTz, PBB, and PbS-EDT films were measured and found to be 107°, 101°, 101°, and 98°, respectively (Figure 4a–c and Appendix A). Interestingly, the EDT treatment of the PbS CQDs drastically decreased their hydrophobicity, which implies poor interfacial contact between the PbS-TBAI and PbS-EDT layers. On the other hand, the CPs showed hydrophobicity similar to the PbS-TBAI layer; therefore, CP/PbS-TBAI layer-by-layer can make better contact at the interfaces.

The surface roughness was analyzed by atomic force microscopy (AFM), as shown in Figure 4d–f. The RMS roughness (*R*_q_) values of the PSBOTz, PBB, and PbS-EDT films on the PbS-TBAI layer were 1.7, 1.1, and 1.2 nm, respectively. All the films showed smooth and uniform surfaces. Although the surface roughness of the PBB film was the lowest among the three films, there is no meaningful difference.

Two-dimensional grazing incidence wide-angle X-ray scattering (2D GIWAXS) images of pristine PSBOTz and PBB films, as shown in Figure 5, were compared to understand the difference in the molecular ordering. Along the *q*_xy_ axis, the two polymers did not show any significant difference in the ordering, but along the *q*_z_ axis, there were some different ordering patterns. In the long-distance ordering area (0.2–1.0 Å^−1^), PSBOTz showed lamellar ordering peaks with clear reflections at (100), (200), and (300), and the corresponding d-spacing was 23 Å, whereas PBB showed clear (100) and (200) reflection peaks with a smaller d-spacing of 21 Å. In the short-distance ordering area (1.0–2.0 Å^−1^), PSBOTz showed two types of π–π stacking ordering at 1.35 and 1.72 Å^−1^, which correspond to a d-spacing of 4.65 and 3.65 Å, respectively, whereas PBB showed a dominant strong π–π stacking at 1.51 Å-1 with a d-spacing of 4.16 Å. Since PBB has a shorter lamellar ordering distance and a more dominant type of π–π stacking ordering than PSBOTz in the film state, PBB could increase the hole mobility and reduce the trap densities in the devices.

## 3. Materials and Methods

### 3.1. Materials Synthesis

Precisely, 3.04 g of lead acetate trihydrate (Pb(Ac)_2_·3H_2_O; Alfa Aesar) and 7 mL of oleic acid (OA; technical grade, Sigma-Aldrich, Seoul, Korea) was dissolved in 40 mL of 1-octadecene (ODE; 90%, Sigma-Aldrich, Seoul, Korea) at 90 °C for 2.5 h under vacuum conditions. The sulfur precursor solution was prepared by adding 0.750 mL of hexamethyldisilathiane ((TMS)_2_S; synthesis grade, Sigma-Aldrich, Seoul, Korea) to 10 mL of ODE. Then, the sulfur precursor solution was injected into the lead precursor solution and reacted for 5 s. After the reaction, 20 mL of toluene was injected into the reactor and cooled to room temperature. The synthesized CQDs were purified by centrifugation with acetone and methanol. The precipitated CQDs were dried overnight under vacuum and dispersed in *n*-octane before device fabrication. The concentrations of the CQD solutions were 80 and 50 mg/mL in *n*-octane for the PbS-TBAI and PbS-EDT layers, respectively. Poly(4-(4,8-bis(5-(2-ethylhexyl)thiophen-2-yl)benzo[1,2-b:4,5-b′]dithiophen-2-yl)-2,6-dioctylbenzo[1,2-d:4,5-d′]bis(oxazole)) (PBB) and poly[(4,8-bis(5-((2-ethylhexyl)thio)thiophen-2-yl)benzo[1,2-b:4,5-b′]dithiophene-2,6-diyl)-alt-((5-bromo-4-octylthiazol-2-yl)thiophene-2,5-diyl)] (PSBOTz) were synthesized according to our synthetic procedures [14,41].

### 3.2. Device Fabrication

Indium-doped tin oxide coated glass (ITO) was used as the bottom transparent electrode. The ITO films were sonicated for 20 min with acetone, distilled water, and isopropyl alcohol. After sonication, the ITO films were stored in a vacuum oven at 80 °C overnight. For the ZnO precursor solution, 0.45 M of zinc acetate dihydrate (Zn(Ac)_2_·2H_2_O, Alfa Aesar, Gongduk-Dong, Mapo-Gu, Korea) and 0.45 M of monoethanolamine (Sigma-Aldrich, Seoul, Korea) were dissolved in 2-methoxyethanol at 60 °C for 3 h. The precursor solution was cooled and aged for 1 d. Before the spin-coating process, the ITO films were subjected to ultraviolet ozone (UV O_3_) treatment for 20 min. The prepared ZnO precursor solution was coated onto ITO at 3000 rpm for 30 s and then thermally annealed at 220 °C for 10 min. The ZnO coating process was repeated once more to obtain a suitable thickness. For the *n*-type CQD layer, the CQD solution was coated onto the ZnO-coated ITO at 2000 rpm for 10 s. Precisely, 20 mM of tetrabutylammonium iodide (TBAI; Sigma-Aldrich, Seoul, Korea) in methanol was used for the ligand exchange process for 30 s and rinsed with methanol twice. The PbS-TBAI layer coating process was repeated five times. The PbS-EDT layer, which was used as the control device, utilized a 50 mg/mL CQD solution and 4 mM of 1,2-ethanedithiol in ACN solution. The PbS-EDT layer coating process was similar to the *n*-type CQD layer and was repeated twice. The polymer HTL solution was prepared by dissolving the polymer in chlorobenzene at a concentration of 5 mg/mL in a N_2_-filled glove box. After stirring overnight, each polymer HTL solution was coated onto the PbS-TBAI layer at 2000 rpm for 40 s and then thermally annealed at 80 °C for 5 min. MoO_X_ and Ag were thermally deposited under high vacuum conditions (<10^−6^ Torr) and acted as the hole-blocking layer and top electrode, respectively. The final active area was calculated as 0.09 cm^2^.

### 3.3. Device Characterization

The absorbance spectra were measured using a UV/Vis spectrometer (Scinco Mega-800, Seoul, Korea). Morphological images were obtained using atomic force microscopy (AFM; SPA400, SII, Chiba, Japan) in the tapping mode. Field emission scanning electron microscopy (FE-SEM; JSM-7610F, JEOL Ltd., Tokyo, Japan) was used to confirm the thickness of each layer in the device. The current density–voltage (*J*–*V*) curve was obtained using a source meter (Keithley 2602B, Cleveland, Ohio, USA). A collimated NIR LED (M940L4, Thorlabs, Newton, N.J. USA) was used to illuminate NIR light in the device. An external quantum efficiency (EQE) spectrometer (QuantX-300, Newport, RI, USA) was used to record the EQE spectrum as a function of the wavelength. The charge carrier mobility was calculated using the space-charge-limited current (SCLC) method with the Mott–Gurney law. The hole-only device (ITO/PEDOT:PSS/PbS-EDT or CP/MoO_x_/Ag) was measured under dark conditions to evaluate the hole mobility.

## 4. Conclusions

We synthesized and utilized two types of CPs, PSBOTz and PBB, to replace the PbS-EDT layer for high-performance QPDs. The HOMO energy levels of PSBOTz, PBB, and PbS-EDT were −5.3, −5.5, and −5.0 eV, respectively, and only PBB showed well-aligned HOMO energy levels in the QPDs. In addition, the PBB polymer exhibited the highest hole mobility of 9.6 × 10^−3^ cm^2^/V·s among the three types of hole-transporting materials. More importantly, the low dielectric constant and insulating properties of the CPs under dark conditions effectively suppressed *J*_d_ in the QPDs. Therefore, the *D** values of the PBB-based devices were four times higher than those of the PbS-EDT-based devices. A significant enhancement in the QPD performance was achieved in the dynamic characteristics. The *f*_−3dB_ of the PBB-based devices was 19.3 kHz, which was one order of magnitude higher than the other devices. The TPV measurements revealed that both *t*_r_ and *t*_f_ were significantly decreased in the PBB-based devices. The sizable molecular ordering and fast hole transport of PBB in the film state, and the low trap density, well-aligned energy levels, and good interfacial contact in the PBB-based devices greatly improved the dynamic characteristics of the QPDs.

## Data Availability

Data are available upon request.

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
