# Peer review of "Improvement of Dynamic Performance and Detectivity in Near-Infrared Colloidal Quantum Dot Photodetectors by Incorporating Conjugated Polymers"

_molecules, 2022, doi:10.3390/molecules27217660_

Round 1

Reviewer 1 Report

Minor comments:

Band gap calculation need to explain in detail (Equation 1); why authors have selected that particular equation since there are many other methods reported?

XRD data should be added with explanation 

Raman spectra analysis data should be presented 

FT-IT of conductive polymer must be added with characterization in details

Although material synthesize done by authors, but there is no proper characterization found, instead of citing reference, clear characterization data must be added

Hole mobility calculation (including equations) should be added in the main manuscript 

There are several typos found, so authors should revise the manuscript carefully

Author Response

Minor comments:

Band gap calculation need to explain in detail (Equation 1); why authors have selected that particular equation since there are many other methods reported?

A) Thank you for your comments. In case of CQDs, the bandgap is tunable depending on the particle size. Thus, we tried to calculate bandgap from the particle size using Moreels equation (Equation 1). TEM images of PbS CQDs (Figure S1b) show the particle size clearly, and the calculated bandgap from the CQD size was 1.4 eV, which is in a good agreement with the optical bandgap of CQDs. According to the reviewer’s comment, we added more explanation about Equation 1. In addition, Equation 1 was moved to Equation 2 during the revision process.

Added sentence in the Page 3

“Since the bandgap (Eg) of the CQDs depends on the particle size, the bandgap can also be calculated by measuring the size of the CQDs (Equation 2) from the transmission electron microscopy (TEM) images of PbS CQDs [41].”

XRD data should be added with explanation 

A) Thank you for your comments. We measured XRD of PbS CQDs and give a proper explanation.

Figure S1. X-ray diffraction spectrum of PbS CQD coated on ITO-patterned glass.

Added sentences in Page 2

“The X-ray diffraction (XRD) spectrum of the synthesized PbS CQDs was measured and shown in Figure S1. The diffraction peaks of PbS CQDs appeared at 25, 31, 42, 51, 61, 69, and 71° which corresponded to (111), (200), (220), (311), (400), (331), and (420) crystal lattices, respectively.”

Raman spectra analysis data should be presented 

A) Thank you for your suggestion. However, there was not enough time for the Raman spectra of the synthesized PbS CQDs. Instead of the Raman spectra, we measured TEM images and XRD spectrums to identify the synthesized PbS CQDs.

FT-IT of conductive polymer must be added with characterization in details

A) FT-IR spectra of synthesized polymers were measured and the characteristic IR peaks are identified as shown in Figure S5.

Figure S5. FT-IR spectra of (a) PSBOTz and (b) PBB conjugated polymers.

Added sentences in Page 3

“FT-IR spectra of PSBOTz and PBB were shown in Figure S5. Both polymers showed C-H, C-C, C-N and C-S stretching and C-H and C-C bending peaks in common, but PBB showed additional strong C-O stretching peak at 1220-1183 cm-1 due to the existence of C-O bonding on the BBO moiety.”

Although material synthesize done by authors, but there is no proper characterization found, instead of citing reference, clear characterization data must be added

A) Thank you for your comments. We added 1H-NMR spectra of PSBOTz and PBB in Figure Sx, and you can also find them in the “Supporting Information” of our two papers. (Polym. Chem., 2019, 10, 4314-4321, Adv. Opt. Mater., 2022, 10, 2102607)

Added sentences in Page 3

their polymeric structures and 1H NMR were recorded in Scheme 1 and Figure S3-4, respectively [43,44].”

Figure S3. 1H NMR of PSBOTz

Figure S4. 1H NMR of PBB

Hole mobility calculation (including equations) should be added in the main manuscript 

A) The hole mobility was calculated via Mott-Gurney’s law using following equation;

Where  and  were the vacuum permittivity and relative dielectric constant, respectively. In this paper, the  value was set as 3.5 and 21.2 for the polymer HTL and PbS-EDT, respectively which were most widely used value. L was the thickness of active layer.

There are several typos found, so authors should revise the manuscript carefully

A) We carefully checked several typos.

Reviewer 2 Report

The paper entitled “Improvement of Dynamic Performance and Detectivity in Near-Infrared Colloidal Quantum Dot Photodetectors by Incorporating Conjugated Polymers” by Kim et al reports the improvement of PbS CQDs based photodetector performance by replacing the conventionally adopted P type CQDs layer by a polymer material allowing better energy alignment and higher holes mobility. The paper is well written, and the interpretation supports the experimental finding. This work comes to complete the recent finding of the authors concerning the development and improvement of organic photodetectors.

I have only two minor suggestions that authors are advised to mention in the discussion part of the text

The authors should, may be, comment on the applicability of their suggested polymer for bigger size PbS CQDs (longer wavelength) other types of pf CQDs such us CdSe, and a wide range of perovskite QDs

It is also recommended to comment on the expected improvements, if any, of organic emitters and solar cells based on CQDs

Author Response

The paper entitled “Improvement of Dynamic Performance and Detectivity in Near-Infrared Colloidal Quantum Dot Photodetectors by Incorporating Conjugated Polymers” by Kim et al reports the improvement of PbS CQDs based photodetector performance by replacing the conventionally adopted P type CQDs layer by a polymer material allowing better energy alignment and higher holes mobility. The paper is well written, and the interpretation supports the experimental finding. This work comes to complete the recent finding of the authors concerning the development and improvement of organic photodetectors.

I have only two minor suggestions that authors are advised to mention in the discussion part of the text

The authors should, may be, comment on the applicability of their suggested polymer for bigger size PbS CQDs (longer wavelength) other types of pf CQDs such us CdSe, and a wide range of perovskite QDs

A) Thank you for your comments. Currently, we are exploring the photodetecting properties at the longer wavelength. We cannot share the data in this stage, but the conjugated polymers are working better in the longer wavelength area. In the NIR and SWIR research, the bottleneck is to make bigger size CQDs with excellent light-detecting properties in the 1000 – 1500 nm.

It is also recommended to comment on the expected improvements, if any, of organic emitters and solar cells based on CQDs

A) Thank you for your comments. Those polymers should work in solar cell application because solar cells are also photodiodes.

Added sentences in Page 2

“In addition, several CPs were successfully utilized as a hole transporting layer in the photovoltaic applications [37, 38].

Reviewer 3 Report

The authors used conjugated polymers as the hole transporting layer to replace the commonly used PbS-EDT layer in quantum dot (QD) based photodetectors. The authors showed that the polymers offer a suitable band alignment, higher hole mobility, and better film uniformity, resulting in photodetectors with higher detectivity and faster response. The authors utilized a range of characterization techniques to measure the materials and electronic properties of the materials and devices; however, using polymers or organic molecules to replace the EDT layer is not a new topic. More evidence should be provided to support the advantages of using polymer HTLs.

The main question here is about the device architecture where MoO3/Ag is used instead of Au. As previous studies have shown, the PbS-EDT layer can form a good interface and Schottky junction with Au, and the employment of MoO3 decreases the stability and fill factor of solar cells due to the unfavorable band alignment (ref. DOI:10.1038/NMAT3984). The authors should explain why MoO3/Ag was selected for this research. A fair comparison could be the polymer/MoO3/Ag and PbS-EDT/Au.

The authors should also compare their device performance with the state-of-the-art QD-based photodetectors at the same detection wavelengths, and explain the performance discrepancy.

Regarding the morphological characterization using AFM, the differences in surface roughness are within the error bar, so it couldn’t be considered as evidence for a smoother film. For the contact angle measurement, the contact angles measured for the polymers are the angle between PbS-TBAI and polymer solution, whereas the control one measures the angle between EDT solution and PbS-OA film. This comparison is not meaningful to me.  

Author Response

The authors used conjugated polymers as the hole transporting layer to replace the commonly used PbS-EDT layer in quantum dot (QD) based photodetectors. The authors showed that the polymers offer a suitable band alignment, higher hole mobility, and better film uniformity, resulting in photodetectors with higher detectivity and faster response. The authors utilized a range of characterization techniques to measure the materials and electronic properties of the materials and devices; however, using polymers or organic molecules to replace the EDT layer is not a new topic. More evidence should be provided to support the advantages of using polymer HTLs.

A) Thank you for your comments. Currently, the development of conjugated polymers gives a great contribution on perovskite and QD solar cells. The tunable HOMO energy levels and high hole mobilities of conjugated polymers are unique advantages on the photovoltaic application as a hole transporting layer. However, in the photodetectors, the effect of conjugated polymers as a hole transporting layer was not well-studied and we investigated QPD performance depending on the conjugated polymers. Finally, we found that the introduction of conjugated polymers are effective to reduce the dark current density and improve the dynamic properties in QPDs.

The main question here is about the device architecture where MoO3/Ag is used instead of Au. As previous studies have shown, the PbS-EDT layer can form a good interface and Schottky junction with Au, and the employment of MoO3 decreases the stability and fill factor of solar cells due to the unfavorable band alignment (ref. DOI:10.1038/NMAT3984). The authors should explain why MoO3/Ag was selected for this research. A fair comparison could be the polymer/MoO3/Ag and PbS-EDT/Au.

A) Thank you for your comment. MoO3/Ag electrode is much cheaper than Au electrode. We think that Au-based devices needs to change to MoO3/Ag electrode in the next step for the commercialization, and thus, we did not set-up to deposit Au in the devices. Please understand our situation. In addition, the comparison between polymer/MoO3/Ag and PbS-EDT/MoO3/Au is fair because we only control the hole transporting layer. Finally, our research plan is to find new conjugated polymers enable to increase photodetecting performance in MoO3/Ag systems.

The authors should also compare their device performance with the state-of-the-art QD-based photodetectors at the same detection wavelengths, and explain the performance discrepancy.

A) Thank you for your comments. This study does not focus on high-performance QPDs. If we focus on getting highest QPD performance, your suggestion is reasonable. However, in this study, we focus on the effect of conjugated polymers on QPD performances, thus the direct comparison between our QPDs and the state-of-the-art QPDs is meaningless. In addition, we already gave some review on NIR QPDs in the “Introduction”. Finally, our key story in this study is that the utilization of conjugated polymers is advantageous to decrease dark current density and improve dynamic properties in QPDs.

Regarding the morphological characterization using AFM, the differences in surface roughness are within the error bar, so it couldn’t be considered as evidence for a smoother film. For the contact angle measurement, the contact angles measured for the polymers are the angle between PbS-TBAI and polymer solution, whereas the control one measures the angle between EDT solution and PbS-OA film. This comparison is not meaningful to me.  

A) Thank you for your comments. We agree with your comments and revised the AFM analysis.

Added sentence in Page 8

“Even though the surface roughness of the PBB film was the lowest among the three films, there is no meaningful differences.”

Round 2

Reviewer 3 Report

The authors claim that the EDT/MoO3/Ag is used because Au electrode should be replaced in the long term and they only compare the hole transporting layer. This explanation is not convincing as the EDT/MoO3/Ag suffers from poor band alignment. This causes carrier trapping and accumulation, and will directly affect the response time of the photodetector. In addition, important evidence provided in this article including the band alignment comparison, detectivity, and response time measurements is all based on the bad control architecture. This makes the comparison unfair. 

Author Response

Dear Editor,

Thank you very much for your letter with regard to our manuscript (Manuscript ID: molecules-1938176) entitled “Improvement of Dynamic Performance and Detectivity in Near-Infrared
Colloidal Quantum Dot Photodetectors by Incorporating Conjugated Polymers
" together with the comment from the Reviewer 3. I appreciate that the reviewer gave us helpful advice and comments. We are sending herewith the revised manuscript. The modified and added sentences are highlighted by “tracking mode”. I am also uploading the cover letter which describes the response to comments.

I hope that our revised manuscript would be suitable for the publication in Molecules.

If you have any question, please feel free to let me know. My phone number is +82-10-2731-2968 and e-mail address is inhjung@hanyang.ac.kr

Thank you for your consideration.

Sincerely,

In Hwan Jung                              

Associate Professor       

Department of Organic and Nano Engineering, Hanyang University

222 Wangsimni-ro, Seongdong-gu, Seoul 04763, Republic of Korea

Tel: 02-2220-0494, E-mail: inhjung@hanyang.ac.kr

----------------------------------------------------------------------------------------

Comments and Suggestions for Authors

The authors claim that the EDT/MoO3/Ag is used because Au electrode should be replaced in the long term and they only compare the hole transporting layer. This explanation is not convincing as the EDT/MoO3/Ag suffers from poor band alignment. This causes carrier trapping and accumulation, and will directly affect the response time of the photodetector. In addition, important evidence provided in this article including the band alignment comparison, detectivity, and response time measurements is all based on the bad control architecture. This makes the comparison unfair. 

  1. Thank you for your comments. According to your advice, we tried to fabricate the PbS-EDT/Au devices, but there is no any difference between PbS-EDT/Au and PbS-EDT/MoOx/Ag devices (Figure R1). It is quite difficult to explain it in this stage because the higher energy level of Au (-5.1 eV) did not improve the photodetecting performances. We expect this to come from suboptimal PbS-EDT/Au devices or still unsatisfactory band alignment. To understand this unusual behavior in PbS-EDT/Au devices, more detailed experiments are required, and we will work on them in the near future. However, since this is a very important point in this study, I clearly point out this band alignment issue in the main text.

Added sentence in Page 3

“To utilize PbS-EDT as a hole transporting layer, it is considered that an alternative electrode capable of better band alignment with the PbS-EDT is needed.”

Figure R1. (a) The expected band alignment of the devices. J–V curves of the device with a structure of (b) PbS-EDT/MoOx/Ag and (c) PbS-EDT/Au devices.

English language and style are fine/minor spell check required

  1. Thank you for your comment. Some typos were corrected in the “tracking mode”.
